# Religious Self-Identification and Culture—About the Role of Religiosity in Cultural Participation

**Magdalena Lipnicka *** and **Tomasz Peciakowski ***

Institute of Sociological Sciences, The John Paul II Catholic University of Lublin, Al. Raclawickie 14, 20-950 Lublin, Poland
* Correspondence: magdalena.lipnicka@kul.pl (M.L.); tomasz.peciakowski@kul.pl (T.P.)

**Abstract:** Culture has its source and anchoring in religion, but the presence of religious values in social life takes place in terms of culture and through culture. Religiosity plays a key role in defining the boundaries of cultural differences, and this paper raises questions as to the extent, ways and environments where religiosity may influence active cultural participation. The research paper attempts to identify the impact of religiosity on such activities. This involves determining which dimensions of religiosity should be distinguished and are most relevant for cultural participation in Poland. The study showed that religiosity influences cultural participation, but ambiguously. Religious self-identification and spiritual self-identification turned out to be the most significant factors. Religiosity is relevant for cultural participation, but mainly when it is a significant element of individual identity.

**Keywords:** religiosity; cultural participation; Poland; religious self-identification; spirituality





## 1. Introduction

Religion and culture are closely intertwined dimensions, and between them, there is an ever-recurring dialectical tension (Świątkiewicz 2020). This is because cultural heritage is expressed in the fusion of cultural and religious themes, and the universe of religious symbols forms the basis of socially objectified and subjectively real cultural meanings. Thus, participating in culture appears as an adaptation to life's realities.

Of course, religion influences culture as well as many other social aspects. McGuire (2002, p. 1) explicitly writes that "religion is one of the most powerful, deeply felt, and influential forces in human society", although this definition raises many issues. Smith's substantive definition (Smith 1995 after: Newman 2004) states that religion is "systems or structures consisting of specific kinds of beliefs and practices: beliefs and practices that are related to superhuman beings" (p. 893). Beyond beliefs and practices, religion is often viewed multidimensionally. Thus, we can speak of aspects or dimensions of religion such as beliefs, rituals, experiences or a community (McGuire 2002, pp. 15–22). Swidler (2014) distinguishes four very similar aspects: Creed (cognitive aspect), Code (behavioral aspect), Cult (ritual aspect) and Community (social aspect).

In 1918, Simmel (2000, p. 89) wrote: "One of the most profound spiritual dilemmas of [ . . . ] modern men is that although it is impossible to preserve the traditional church religions any longer, the religious impulse still exists". This diagnosis seems to be valid, because presently, the difference and distance between institutionalized religion and the meaning that an individual ascribes to religion in one's life are more noticeable (Motak 2010). Thus, individual, personal religiosity loosens or completely breaks with traditional religious institutions, and people are looking for new spaces that are more adequate for modern societies. Therefore, Simmel (2000) presented the development of culture as a constant search for new forms.

Religious and spiritual transformations take place not only outside churches, but also inside churches (between orthodoxy and heterodoxy). Relationships with traditional

religious institutions are weakening, but a religious element is still needed as emotional support for the community and as a factor ensuring the identity of an individual and society. This is related to the concept of religiosity as the personal and communal expression of people's connections to a particular religion. It allows us to see the contemporary changes from institutionalized religion to individualized religion, in which individuals, rather than institutions, and individual experience, rather than institutionalized patterns of behavior, are becoming more important. This "turning to experience" noticeable in religiosity reflects the spirit of the times. Bauman (2000) wrote about focusing on "experience" as the main characteristic of the liquid phase of modernity, reflecting society's individualism and consumption trends. Individualization, and with it secularization, imply that religious experience depends less and less on belonging, and more and more on individual choice. It is also worth noting that secularization today is not about the disappearance of religion, but about the emergence of a new spirituality, new forms of religiosity. An institutionally specialized form of religion is only one among many other choices (Berzano and Sheridan 2019).

Some researchers (Stark and Glock 1965) mention the consequences of religion, which admittedly do not concern religion per se, but how religion influences other spheres of life. It is worth emphasizing that religiosity is entangled in various social processes; on the one hand, culture has its source and anchoring in religion, yet, on the other hand, the presence of religious values in social life patterns takes place in terms of culture and through culture (Świątkiewicz 2020). Katz-Gerro and Jaeger (2012) wrote that religiosity plays a key role in defining the boundaries of cultural differences, and if so, the question arises as to how much, in what ways and the environments where religiosity may influence active cultural participation. This paper attempts to identify the impact of religiosity on such activities. The article's contribution to the literature is twofold. First, we will examine which of the identified dimensions of religiosity is most relevant for active cultural participation. Second, we will determine the relevant characteristics and variables that influence cultural participation, showing to what extent these influences change the introduced interactions with religiosity. The analyzes are based on the results of the research from a CATI survey conducted in September 2020 in Poland on a nationwide, random and representative sample of 1010 Poles (in the 15+ age category).

The article is organized as follows: first, we will review the literature on cultural participation and religiosity. Second, we will build a model of religiosity (distinguishing four aspects). Third, we will describe the results. The article ends with a discussion in which we also refer to the cultural context of the studied population and present proposals for further research.

## 2. Literature Review

### A.    Cultural participation

The terms 'cultural participation' and 'cultural consumption' (as well as a 'participant' and 'consumer' of culture) are often used interchangeably. It needs to be emphasized that the concept of participation in culture is conceptually somewhat different from the concept of behaviors commonly included in cultural consumption. Consumption is a one-dimensional concept, focusing on processing or the reception of cultural products. In the broad sense, it is more of "the act of using" (Cambridge Dictionary 2021), the reception of culture in general. Presently, it is increasingly difficult to set limits to cultural consumption, because we permanently consume it.

The concept of participation is an attempt to emphasize the importance of symbolic interactions and meanings (communicative function). Next, it is the system of values, norms and patterns (axiological function) and finally, the exchange of information, codes and interaction with another person (social function) (Morrone 2006). Therefore, cultural participation requires activity, and it refers to "the fact that you take part or become involved in something" (Cambridge Dictionary 2021). This can be understood in a broader sense as an individual participating in cultural phenomena, or in a narrower sense as participating

in artistic culture (where creative or receptive participation can be distinguished). We have assumed that participating in cultural events (going to the cinema, theater, concerts, meetings with artists, celebrations, etc.) is a type of declared cultural activity.

Cultural participation is conditioned by many factors. One can point to explanations through emotional, intellectual as well as social factors (Manolika and Baltzis 2020). Due to social factors, the problem of cultural participation is a topic of interest for sociologists. The most important and widespread theoretical perspectives that can be pointed out regarding the study of patterns of cultural participation are those of Pierre Bourdieu (1979). In *La Distinction. Critique sociale du jugement*, he reveals the historical and social genesis of aesthetic taste and its connection with the social structure, economics and cultural capital (Kisiel 2013, p. 346). The second important research trend referring to and developing Bourdieu's theory is presented by R. Peterson (1992), who introduced the univorous-omnivorous category still used by researchers around the world.

In addition to associating cultural participation with social structure and capitals (e.g., Bennett et al. 2008; Bryson 1996; Holt 1998), other factors such as one's family situation, attitude towards art or professional status are also indicated (DiMaggio 1996; Smith and Wolf 1996; Kolb 2002; Ostrower 2005; Swanson et al. 2008). Some studies have also pointed to gender (Christin 2012), issues of social and relational ties (Upright 2004), education and country of origin (Falk and Katz-Gerro 2016) or ethnicity (van Wel et al. 2006) as factors influencing patterns of cultural participation. A factor that has so far received less attention in research is religiosity, and it requires a few more explanations.

B.　　The meaning of religiosity

Religiosity is the personal and communal expression of someone's ties to a particular religion. The term 'religiosity' is often used interchangeably with terms such as 'spirituality' or 'faith', which leads to many misunderstandings (Jones 2018). Spirituality may or may not be related to religiosity. Religiosity is organizational and concerns beliefs, practices and traditions. Spirituality is abstract, subjective and connected to something beyond ourselves, beyond the physical (Paul Victor and Treschuk 2019). Spirituality is linked to four aspects: the idea of the inner self (Kale 2004; Lun and Wai 2015), the search for meaning in life (Kale 2004; Myers 1990; Lun and Wai 2015), connection to others (Kale 2004; Lewis et al. 2007), and connection to the Absolute (however, it may be defined religiously or non-religiously) (Kale 2004; Lun and Wai 2015). Although subjective spirituality is in a sense opposed to institutional religiosity (Van Niekerk 2018), spirituality and religiosity can intermingle (I am religious and spiritual), yet they can be separated (I am not religious, but I am spiritual or vice versa).

Faith, on the other hand, is an even more ambiguous and difficult concept to define. Newman (2004) notes that it is an act of will, not intellect, and can refer to "a general religious attitude on the one hand to personal acceptance of a specific set of beliefs on the other hand" (Hellwig 1990, p. 3 after Newman 2004). Newman (2004) proposed three categories: knowing (faith), doing (religious) and being (spiritual). According to Newman's model, faith is the basis for religiosity and spirituality (which seems to be understood by Newman as inner convictions). "In contrast to this common approach, in my model, spirituality and religion are functions of faith. Both religion and spirituality require faith as a foundation (...). In other words, faith is the guiding principle by which individuals are either religious or spiritual" (Newman 2004, p. 106). Worth noting here, however, is that although Newman makes a distinction between being and doing, these can interpenetrate. A good example is Buddhism, where doing can also mean being (spiritual). These two may or may not be combined.

In light of the above, four cross terms will apply in our model: belief (in the sense of declared faith), practices (in the sense of undertaken religious rituals), *religious self-identification* (in the sense of identification with an institutional religion) and *spirituality* (leading a spiritual life, caring for one's spiritual development).

Including the aspects of spirituality and religiosity in self-identification is important due to the processes of secularization and desecularization. The absence of these aspects in

the analysis could mean a loss of insight concerning changes in contemporary religiosity. There is no actual decline in the importance of religiosity as previously thought, but there is a change in its meaning, a detraditionalization and a shift to a more individual and private sphere (Weisse 2010; Gunnarson et al. 2016; Mariański 2004). In contemporary Europe (both Western and Eastern, though to varying degrees), we can speak of the process of the individualization of religion, which also translates into the process of secularization, mostly according to Taylor's (1992) rather than Berger's (1967) theory. Religion moves into the private sphere, faith and religious practices have a lesser influence on other areas of an individual's life, as they become a personal matter. This may or may not lead to a decline in religion.

Nevertheless, the social aspect of religion has not lost its importance. Religion is still associated with a group of followers belonging to a particular religion. For example, identifying yourself as a follower of a certain religion can give you a sense of belonging, sharing experiences, participating in rituals and a sense of community. This is directly related to social ties, a network of contacts, etc. in a particular religious group (McGuire 2002). Above all, religiosity becomes an important element of an individual's identity.

C.    The impact of religiosity on cultural participation

It has been confirmed many times in social sciences that religion and religiosity affect various spheres of social life. These spheres include: (a) political beliefs (Sherkat and Ellison 1999; van Eijck and Bargeman 2004; Radu 2010); (b) family relationships (Horwath et al. 2012; Sherkat and Ellison 1999; Krok 2016); (c) health and well-being (Sherkat and Ellison 1999; Yeniaras and Akarsu 2017; Bertelli et al. 2020); and (d) social capital (Sherkat and Ellison 1999). Furthermore, religiosity (in its various dimensions) can influence consumer and economic behavior, such as fashion trends (Farrag and Hassan 2015), entrepreneurship (Chebotarov and Chebotarov 2020) and some consumption behaviors (e.g., Nayar et al. 2010). The topic of the influence of religiosity on different aspects of people's lives and decisions is very broad and also depends on the extent of religiosity.

In the literature (Katz-Gerro 1999; Katz-Gerro and Jaeger 2012; Greeley 2001; Van Eijck 2011; Montoro-Pons and Cuadrado-García 2015) on the impact of religiosity on cultural participation and cultural consumption, religiosity affects cultural participation and cultural consumption. An article by Katz-Gerro (1999) includes that 'going to church' is one of the variables that correlates positively with participating in high culture. Tally Katz-Gerro and Mads Meier Katz-Gerro and Jaeger (2012) explicitly point out that there is a positive correlation between religiosity and cultural consumption (and that this influence is comparable to that of important socio-economic factors such as education and socio-economic status).

At the same time, the authors point out that there is little evidence that denomination is significantly related to cultural consumption (which is also confirmed by the studies indicated above). The authors link the above results to the activities of religious people, such as participating in religious practices, social networks (social capital) and the role of religion in determining social divisions. It is usually assumed that religion indirectly influences one's social status, privileges and prestige, thus affecting cultural consumption, but the study by Katz-Gerro and Jaeger (2012) does not confirm this. "The influence that religion has on individuals' access to power, privilege, and prestige can be restudied in the light of our robust cross-national finding that an association exists between religion and consumption" (p. 356).

A second important study that cannot be ignored is the article *Religiosity and cultural consumption* (Montoro-Pons and Cuadrado-García 2015). The authors identify two aspects through which religiosity can influence consumption and cultural participation. First, religiosity is understood in the functional sense as related to group membership and social relations. This concerns the religious participation effect, which consists in the fact that we benefit from participating in religious social networks. Second, it happens through the degree of one's attachment to beliefs and convictions. In this sense, religiosity can strongly influence ethical convictions, also towards one's culture (in a chosen content)

and individual decisions regarding participation in certain cultural events, etc. Based on empirical data from the USA, the authors concluded that the influence of religiosity is ambiguous and depends on one's point of view. The first aspect increases the probability of cultural participation due to social and network aspects. Yet, it is also pointed out that greater involvement in religious practices may leave less time for other cultural activities, while the second aspect, beliefs, decreases this probability.

In the literature, we find a justification for the study of the relationship between religiosity and cultural participation. Religiosity, as we have tried to show above, is not a simple phenomenon and should not be reduced to it. It can influence cultural participation, but not necessarily in all its dimensions. Therefore, this article will begin by checking how particular dimensions of religiosity influence cultural participation. Based on the above assumptions, our first hypothesis states that:

**Hypothesis 1 (H1).** *Religiosity affects cultural participation in different ways, depending on the dimensions of one's religiosity.*

Nowadays, modernization processes associate social changes with the transition from a "world of fate" to a "world of choice," from imposed social norms to individual freedom, from absolute imperatives to relatively unlimited possible options (Mariański 2019; Berger 1979). This means that contemporary culture deals with offers and proposals, not orders and standards (Bauman 2000). Individualization affects symbolic transformations at the level of values, ideas and cultural patterns, which also includes religious and moral norms. Religious individualization means that an individual becomes the ultimate authority in matters of faith and morals, and not one's church institution as in the past. Luckmann's (1967) "invisible religion" is a non-institutionalized or not fully institutionalized religion, distant from a public ritual; it is religiosity not only at the individual level (including small communities or "secondary" institutions), but one's identity becomes the last instance of what is religious.

More and more often one writes about the renaissance or return of religion, resacralization or a post-secular society (Mariański 2017). It's members behave like consumers, satisfying their needs by choosing from among a rich offer of religious or secular products on the free market. Today, building an identity is choosing from all available options. Anthony Giddens confirms that:

> Under the conditions of late modernity, everyone has a lifestyle and, in addition, is forced to do so in a significant sense: there is no choice–you have to choose. A lifestyle can be defined as a more or less integrated set of practices adopted by an individual, not only because they are useful, but also because they give a material shape to individual identity narratives. (Giddens 1991, p. 81)

Changes in culture cause changes in the forms of religiosity, and this shapes one's personal identity and its external presence in the form of practices. Therefore, if religiosity influences cultural participation, it is not because of the religious practices undertaken by an individual, not because she or he believes or does not believe in God, but because religious values and norms are an important part of an individual's identity, and so people need to develop their spirituality. Therefore, we put forward another hypothesis:

**Hypothesis 2 (H2).** *Cultural participation is influenced by the dimensions of religiosity that are mostly related to self-identification.*

Basic socio-demographic characteristics are of great importance for individual identity. Hence, we must look at the relationship examined in this context. Studies on the impact of religiosity on cultural consumption and participation can be complex and take into account other variables such as gender (women are more often religious), age (positive correlation), and place of residence (rural residents are more religious) (cf. Iyer 2016). Research by Becker and Woessmann (2013) also shows that there is a link between participating in

religious practices and income. Chan (2019) in his research also notes that, besides religion, education can also influence cultural consumption. Thus, there is a correlation between occasional participation in religious practices and cultural consumption.

Research shows that cultural participation is highest in metropolitan societies, that it increases with education and decreases with age (GUS–Główny Urząd Statystyczny 2020). These conclusions are also confirmed by the research results presented below. Earlier hypotheses indicated, however, that religiosity positively influences cultural participation, but only in the dimensions relating to individual self-identification. Continuing the line of thought proposed in H1 and H2, one can speculate that in social categories with low cultural participation, meaning religiosity in the dimension of self-identification, should also positively influence cultural activities. Therefore, religious self-identification increases religious participation among the elderly (over 65), people from small towns and people who completed their elementary and secondary education, thus offsetting the impact of demographic characteristics on cultural participation in these categories. The third hypothesis states:

**Hypothesis 3 (H3).** *Religiosity influences key demographic characteristics of cultural participation by influencing the cultural participation of social groups that tend to participate less in culture.*

### 3. Materials and Methods

The data comes from a survey conducted in September 2020 on a nationwide, random and representative sample of 1010 Poles, aged 15+ (Table 1). It was conducted using the CATI method based on an interview questionnaire. Due to the ongoing Covid-19 pandemic, respondents were asked about cultural activities undertaken before the pandemic period (before lockdown, closure of cinemas, theaters and various cultural institutions). Below we present basic statistics defining the structure of the research sample.

**Table 1.** The results of summary statistics.

| Variables | Category | N/M | %/SD |
|---|---|---|---|
| Cultural Participation | No | 313 | 31.0 |
| | Yes | 697 | 69.0 |
| Sex | Man | 483 | 47.8 |
| | Woman | 527 | 52.2 |
| Education | Primary | 175 | 17.3 |
| | Vocational | 240 | 23.8 |
| | Secondary | 347 | 34.4 |
| | Higher | 248 | 24.5 |
| Place of residence | Rural area | 398 | 39.4 |
| | Town < 20,000 | 133 | 13.1 |
| | Town 20–99,000 | 195 | 19.3 |
| | City 100–499,000 | 168 | 16.6 |
| | City > 500,000 | 117 | 11.6 |
| Age | Min. 15 Max. 86 | 46.74 | 17.550 |

For the analyses, it is crucial to establish the indicators explaining cultural participation and also those explaining religiosity. In our model, we will refer to four dimensions of religiosity: Belief (B), Religious Practices (RP), Religious Self-Identification (RSI) and Spirituality (S). These are tested as explanatory variables along with eight categorized possibilities: believer/non-believer; practitioner/non-practitioner; religious/non-religious; spiritual/non-spiritual. Each of these terms can occur together (I am religious, spiritual, believing and practicing), separately or in a mixed way (e.g., I am believing but not practicing; I am practicing but not religious, etc.). This is only seemingly contradictory, since disbelief means the rejection of belief in a personal God, while non-religiousness means rejecting beliefs associated with a particular religion (Tyrała 2015). Also, declaring

faith or participating in religious rituals does not necessarily have to be associated with religious identification on an individual basis. It may be the result of strongly rooted religious beliefs in social consciousness or the treatment of certain religious rituals as important cultural practices forming a universal social order.

　　　Due to the theoretically complex four-dimensional model of religiosity, for a more precise analysis, we assumed that we would code each dimension of religiosity as a zero-one variable (taking the value of 1 when a phenomenon occurs and 0 when it does not). Reducing it to a dichotomy may be questionable in the case of the religious practices dimension, due to the degree of this feature's variable. Yet, alternative analytical models using an ordinal scale for the RP variable have shown that this insignificantly affects the correlation of this variable. The following tables show the basic statistics for the variables defining the different dimensions of religiosity (Table 2) and the structural relationship between them (Table 3)—all correlations between the different dimensions of religiosity are significant at the $p < 0.001$ level.

**Table 2.** The results of summary statistics for religiosity.

| Variables | Category | N | % |
|---|---|---|---|
| [B]Belief | Yes | 937 | 92.7 |
| | No | 73 | 7.3 |
| [RP]Religious Practises | Yes | 686 | 67.9 |
| | No | 324 | 32.1 |
| [RSI]Religious Self-Identification | Yes | 628 | 70.0 |
| | No | 269 | 30.0 |
| [S]Spirituality | Yes | 410 | 45.7 |
| | No | 487 | 54.3 |

**Table 3.** Relationships among the dimensions of religiosity.

| Variables | % [B] Believers N = 937 | % [RP] Practicing N = 686 | % [RSI] Religious N = 628 | % [S] Spiritual N = 410 |
|---|---|---|---|---|
| [B]Believers | | 100.0 | 98.8 | 97.1 |
| [RP]Practicing | 73.2 | | 85.0 | 79.3 |
| [RSI]Religious | 74.6 | 85.7 | | 79.7 |
| [S]Spiritual | 47.8 | 52.1 | 52.0 | |

[B] = Belief; [RP] = Religious Practises; [RSI] = Religious Self-Identification; [S] = Spirituality.

　　　The set of variables defining the cultural participation index (as the key variable relating to the phenomenon) consists of 7 items (Table 4). It includes: types of practices going to the cinema; watching a live theater performance; attending music events or concerts; participation in ceremonies commemorating local events and people; participation in meetings with cultural personalities (artists); participating in cultural family celebrations, playing music, singing together, etc. The participation rate was determined by asking which of the listed cultural practices the respondent had performed in the past year. Indicating at least one activity was coded as "1" while not indicating was coded as "0".

**Table 4.** Table of dependencies.

|  | B | RP | RSI | S |
|---|---|---|---|---|
|  | $\chi^2$ | $\chi^2$ | $\chi^2$ | $\chi^2$ |
| Cultural participation index | 0.066 | 2.302 | 7.646 ** | 12.609 *** |
| Type of practice: going to the cinema | 3.277 | 9.294 * | 3.615 | 0.434 |
| Type of practice: watching a live theater performance | 0.000 | 0.435 | 10.021 ** | 0.375 |
| Type of practice: attending music events or concerts | 11.027 *** | 0.859 | 0.275 | 0.708 |
| Type of practice: participation in ceremonies commemorating local events and people | 3.808 | 2.689 | 12.701 *** | 14.834 *** |
| Type of practice: participation in meetings with cultural personalities | 0.003 | 0.238 | 5.130 * | 4.823 * |
| Type of practice: participation in cultural family celebrations, playing music, singing together, etc. | 1.966 | 7.833 ** | 14.981 *** | 10.189 *** |
| Other culture variables: |  |  |  |  |
| Diversity of practices | 1.605 | 3.987 | 8.245 * | 15.101 ** |
| Caring for local heritage | 5.694 | 20.883 *** | 18.614 ** | 37.811 *** |
| Family as a participation motivator | 0.610 | 4.066 * | 13.097 *** | 4.665 * |
| Friends as participation motivators | 0.565 | 1.782 | 0.322 | 2.001 |
| School as a participation motivator | 1.380 | 0.311 | 0.042 | 0.81 |
| Church community as a participation motivator | 4.029 | 16.285 *** | 34.367 *** | 18.559 *** |
| Local cultural institutions as participation motivators | 1.009 | 9.043 ** | 8.516 ** | 4.499 * |

* $p < 0.05$; ** $p < 0.01$; *** $p < 0.001$.

Using the chi-square independence test, the relationships between the participation rate, the seven types of cultural practices and the four dimensions of religiosity were analyzed. The analysis was intended to indicate whether the individual dimensions of religiosity interact with cultural participation and, if so, which of these dimensions is most important (defined by the number of statistically significant interactions) for cultural activities.

The key dependent variable, cultural participation rate, is dichotomous [1—present; 0—absent]. Due to the Binomial Logistic Regressio, the method of highest Likelihood Ratio [LK] was applied. This method is typically used by researchers for three reasons: "1. To predict the probability that the outcome or response variable equals 1; 2. To categorize outcomes or predictions; 3. To access the odds or risk associated with model predictors" (Hilbe 2015, p. ix). Our analysis identified the relevant characteristics that interact with the variable, and then we indicate whether these interactions changed based on the introduced interactions and one of the dimensions of religiosity. This objective was achieved in three stages. The first model selected variables that interact with cultural participation. The second model tested the main effects of the selected variables and their impact on participation. Further models show the individual main effects of religiosity, the main effects of demographics and what happens to the impact of demographic variables when interacting with religiosity. More precisely, this means that we consider religiosity in the dimension that previously proved to be the most significant.

## 4. Results

The first part of the analysis, the chi-square independence test, showed that only two dimensions obtained a statistically significant relationship with the index: Religious Self-Identity [RSI] and Spirituality [S]. The highest number of correlations, four, was recorded for the variable Religious Self-Identity [RSI] according to types of cultural practices (with a min. $p < 0.05$), of which two were significant <0.001 (Spirituality [S] received 3 and 2; Religious Practices [RP]—2 and 0). The tests showed only one statistically significant interaction for one of the dimensions of religiosity defined as Belief[B].

In the first stage, all dimensions of religiosity and the four most important (independent of respondent opinions) socio-demographic characteristics such as gender, education, place of residence and age were selected to build the Binomial Logistic Regressio model. The constructed model (Overall Model Test: $R^2N = 0.24$; $\chi^2 = 165.13$; $p < 0.001$) selects the variables defining the individual dimensions of religiosity (Table 5) in terms of their impact on participation. It confirms the results of the chi-square analyses: in fact, the same two dimensions of religiosity turn out to be important, and again the strongest predictor is Religious Self-Identification ($p < 0.001$). Among the socio-demographic variables, gender can be considered insignificant.

**Table 5.** Omnibus Likelihood Ratio Tests.

| Predictor | $\chi^2$ | df | $p$ |
|---|---|---|---|
| B | 0.3 | 1 | 0.585 |
| RP | 2.92 | 1 | 0.087 |
| RSI | 14.55 | 1 | <0.001 |
| S | 5.6 | 1 | 0.018 |
| Gender | 0.09 | 1 | 0.768 |
| Education | 53.06 | 3 | <0.001 |
| Place of residence | 20.58 | 4 | <0.001 |
| Age | 33.83 | 5 | <0.001 |

Through further analysis, the main effects were identified through the values of variables that are associated with cultural participation (Table 6). Both dimensions of religiosity proved to be significant predictors, with a higher odds ratio indicated for being religious [RSI] (OR = 1.887, at $p < 0.001$). It almost doubled the likelihood of cultural participation (being spiritual [S] OR = 1.458; $p = 0.024$). Significant characteristics positively influencing cultural participation are also indicated among the socio-demographic variables. The odds ratio increased with more education, as in secondary education (OR = 3.919, $p < 0.001$) or higher education (OR = 5.94, $p < 0.001$), where it was also true for the largest cities of 500,000 inhabitants or more (OR = 4.725, $p < 0.001$). The age variable, on the other hand, showed that the odds ratio decreases with age, meaning that older people are less likely to participate in culture.

In the final part of the analysis, we decided to test the impact of selected socio-demographic variables and the simultaneous occurrence of interaction on the Religious Self-Identity variable. It was the dimension of religiosity that had the greatest impact on cultural participation (Table 7). Interaction with religiosity reversed the effects on the explanatory variable, and a significant effect with a positive odds ratio was observed in the categories with opposite results based on primary education, inhabitants of small towns below 20,000 and the category of senior citizens over 65 (Table 8). Thus, the analysis showed that when testing the interaction of religiosity and socio-demographic variables as predictors of cultural participation, the highest degree of religiosity was noted in categories that previously had the weakest impact on cultural participation. This dimension of religiosity not only offsets but also reverses the effects of the three tested socio-demographic characteristics.

**Table 6.** Model Coefficients–Cultural Participation.

| Predictor | SE | Z | Odds Ratio | Lower | Upper |
|---|---|---|---|---|---|
| Religious Self-Identity: 1–0 | 0.177 | 3.5848 | 1.887 *** | 1.3336 | 2.671 |
| Spirituality:1–0 | 0.166 | 2.2651 | 1.458 * | 1.0521 | 2.02 |
| Education: | | | | | |
| Vocational–Primary | 0.258 | 1.5705 | 1.499 | 0.9045 | 2.484 |
| Secondary–Primary | 0.265 | 5.1606 | 3.919 *** | 2.333 | 6.584 |
| Higher–Primary | 0.335 | 5.3108 | 5.94 *** | 3.0777 | 11.465 |
| Place of residence: | | | | | |
| Town < 20,000–rural area | 0.244 | 0.4928 | 1.128 | 0.699 | 1.82 |
| Town 20–99,000–rural area | 0.213 | −0.242 | 0.95 | 0.6258 | 1.441 |
| City 100–499,000–rural area | 0.226 | −0.09 | 0.98 | 0.6296 | 1.525 |
| City over 500,000–rural area | 0.422 | 3.6798 | 4.725 *** | 2.0664 | 10.806 |
| Age: | | | | | |
| 25–34–up to 24 | 0.373 | −1.931 | 0.486 | 0.234 | 1.011 |
| 35–44–up to 24 | 0.359 | −2.591 | 0.395 * | 0.1954 | 0.797 |
| 45–54–up to 24 | 0.36 | −3.606 | 0.273 *** | 0.1349 | 0.553 |
| 55–64–up to 24 | 0.353 | −4.262 | 0.222 *** | 0.1114 | 0.444 |
| 65 and more–up to 24 | 0.339 | −4.864 | 0.192 *** | 0.0987 | 0.373 |

* $p < 0.05$; *** $p < 0.001$.

**Table 7.** Omnibus Likelihood Ratio Tests.

| Predictor | $\chi^2$ | df | $p$ |
|---|---|---|---|
| RSI | 7 | 1 | 0.008 |
| Education | 37.69 | 3 | <0.001 |
| RSI*Education | 9.06 | 3 | 0.029 |
| Place of residence | 15.48 | 4 | 0.004 |
| RSI | 3.47 | 1 | 0.063 |
| RSI*Place of residence | 9.07 | 4 | 0.059 |
| RSI | 6.59 | 1 | 0.01 |
| Age | 21.53 | 5 | <0.001 |
| RSI*Age | 4.19 | 5 | 0.522 |

**Table 8.** Model Coefficients-Cultural Participation (table and interactions).

| Predictor | SE | Z | Odds Ratio | Lower | Upper |
|---|---|---|---|---|---|
| RSI*Education: | | | | | |
| Vocational–Primary | 0.589 | 1.9236 | 3.1027 * | 0.9788 | 9.835 |
| Secondary–Primary | 0.581 | 0.9778 | 1.7657 | 0.5649 | 5.519 |
| Higher–Primary | 0.739 | −0.1852 | 0.8721 | 0.205 | 3.71 |
| RSI*Place of residence: | | | | | |
| Town < 20,000–rural area | 0.568 | 2.7957 | 4.8918 ** | 1.6073 | 14.888 |
| Town 20–99,000–rural area | 0.462 | 1.1359 | 1.6894 | 0.6835 | 4.176 |
| City 100–499,000–rural area | 0.476 | 1.9769 | 2.56 * | 1.0081 | 6.501 |
| City over 500,000–rural area | 1.189 | −0.0876 | 0.9011 | 0.0876 | 9.272 |
| RSI*Age: | | | | | |
| 25–34–up to 24 | 0.761 | 0.8408 | 1.8963 | 0.4267 | 8.428 |
| 35–44–up to 24 | 0.735 | −0.2674 | 0.8215 | 0.1945 | 3.47 |
| 45–54–up to 24 | 0.741 | 0.7369 | 1.7269 | 0.4038 | 7.385 |
| 55–64–up to 24 | 0.75 | 0.1917 | 1.1547 | 0.2655 | 5.023 |
| 65 and over–up to 24 | 0.761 | 1.4289 | 2.9664 * | 0.6676 | 13.181 |

* $p < 0.05$; ** $p < 0.01$.

## 5. Discussion

The study confirmed 3 previously assumed hypotheses and showed that religiosity influences cultural participation, but ambiguously. No significant relationship was noted for any of the four dimensions of religiosity adopted in this study. The dimensions related

to religious self-identification (defining oneself as a religious person, self-identification with a particular religion [RSI]) and spiritual self-identification (defining oneself as a spiritual person, caring for spiritual development [S]) turned out to be the most significant areas. This confirms the findings of the cited research (Montoro-Pons and Cuadrado-García 2015) that aspects related to membership and identifying with a religious group influence cultural participation. On the other hand, religiosity is relevant for cultural participation, but mainly when it is a significant element of individual identity. Religiosity reveals its meaning in the construction of identity, both at the community (belonging) and individual levels (spirituality and identifying with specific values). Our sense of belonging and identifying ourselves as a "religious person" and/or a "spiritual person" is much more important than the statistically insignificant meaning of "I am a believer" and/or "I practice my faith".

Of course, we have known for a long time that religion is one of the main factors determining social distinctiveness. It sets social patterns and creates a system of beliefs, the important elements of personal identity. This is also visible in the sphere of the *sacrum*, which upholds certain lifestyles, hierarchies of values, and symbols of group identity accepted by a significant part of society. Integrating individual identification in an intersubjective community often takes place through religion (Casanova 2011). Therefore, the results (Table 4) demonstrated that people's cultural participation in religion is oriented towards local cultural heritage and cultural family events. Being a religious person increases cultural participation in celebrations commemorating historical events, local traditions and customs. Also, being religious increases a person's chances of participating in family celebrations, often related to religious festivals. One's family is also a much greater motivator for someone to participate in cultural events.

As evident in the research, religiosity (mainly speaking, religious self-identification) offsets the influence of key demographic characteristics on cultural participation by influencing cultural participation in the social categories that tend to engage less in culture. This is particularly evident in three aspects: education, place of residence and age. In these three cases, religiosity reverses the results, yet compensates for it in some way. Seniors, people from rural areas and those with only a primary education, participate less in cultural events. However, when religiosity is taken into account, these people get much better results for cultural participation. This is very interesting, because it seems that we can explain it partly by their involvement in religious networks and the local church, which also has a cultural function. One's religious community turns out to be a strong motivator for cultural participation. Thus, one's church also becomes a cultural institution, especially in rural areas and among older people, who usually participate less in cultural events.

This is probably why religiosity has a positive effect on cultural participation among people who, according to research, usually participate less in culture and are less involved in culture. Yet, interestingly, in the presented research but unlike Katz-Gerro (1999), undertaking religious practices and their frequency turned out to be irrelevant concerning cultural participation. Just as insignificant was religiousness related to faith (we will raise this topic later). It seems that is dimension refers more to traditionally understood religiousness, and it undergoes constant changes. Postmodern man satisfies religious needs in various ways, not always through institutionalized religions and Churches. This is rather done in the sphere of non-church religiousness, which is transferred to a private sphere, often a compilation (conglomeration) of elements of various religious and philosophical systems. Secularization processes are indeed taking place in the modern world. Yet, as Giddens (1991) points out, religion is not only disappearing, but it is also experiencing a rebirth. In many postmodern societies with a high degree of secularization, tendencies are pointing to the "spiritual productivity" of the present culture. We are witnessing the emergence of new forms of religious and spiritual sensitivity (Mariański 2019).

Secularization manifests itself not so much in a departure from God as in a search for God on one's own, drawing on the many inspirations available. This consists in going beyond the teachings and catechism of the Catholic Church to adopt, within the

framework of freedom of choice, only those elements that seem justified for the needs of a privatized religious identity (Libiszowska-Żółtkowska 2004). In the dynamic reality of ever-changing patterns of social life, religious identification is now a correlate of cultural phenomena, because "culture is the costume in which religion is dressed and in which religion dresses itself" (Świątkiewicz 2020, p. 53). Perhaps this is why religious and spiritual self-identification influences patterns of participation in culture, the rejection of religious practices and being a believer. Yet even here the specificity of the population may also play an important role.

There is no statistically significant impact of the Belief [B] dimension on cultural participation. This seems to be the limitation of the presented research. According to data from GUS (the Central Statistical Office), almost 94% of Poles aged 16 and over declare that they belong to a religious denomination (91.9% are members of the Roman Catholic Church). However, only 81 percent consider themselves to be believers, only 50 percent declare regular weekly participation in religious practices, another 17 percent practice at least once a month, and 26 percent only on holidays (GUS–Główny Urząd Statystyczny 2018). Furthermore, the *dominicantes* and *communicantes* ratios for the total number of obliged persons represent 38.2% and 17.3% of the total population respectively (ISKK–Instytut Statystyczny Kościoła Katolickiego w Polsce [Institute for Catholic Church Statistics In Poland] 2019). The attitude of Poles varies according to age (older people are more often religious, while the youth are not). Due to the high degree of practicing religion and the important role the Church has played in its history, Poland is often referred to as a Catholic country. However, faith, often tacitly assumed and treated as the essence of religion, is such a specific personality trait that it cannot be reduced to knowledge or acceptance of the Church's teaching, nor even to a declaration of belonging to the Church (Motak 2010). Often, in the context of Polish society, defining oneself as a believer is accepted as an element of Polish culture. Yet, stating so does not necessarily entail real participation in one's religious life. It does not even guarantee that someone follows the moral principles and precepts of a given religion (which applies more to describing yourself as a religious person).

Since the research was conducted in one population (which presents a certain limitation when stating these results as universal conclusions), worth checking are the hypotheses and results in other populations. Interesting questions concern, above all, how religiosity influences cultural participation in an individual way (self-identification), and whether religiosity influences lower participation in culture in other populations as well. This will allow us to answer the question of whether these results are typical for the Polish population, or if they are more universal. The presented results justify the postulate of further research in the field of relationships between religiosity and cultural participation. In the context of the individualization of religion, this relationship may become more and more important in the process of building individual identity.

**Author Contributions:** Conceptualization, M.L. and T.P.; methodology, M.L. and T.P.; software, T.P.; validation, T.P.; formal analysis, T.P.; investigation, M.L. and T.P.; resources, M.L. and T.P.; data curation, T.P.; writing—original draft preparation, M.L. and T.P.; writing—review and editing, M.L. and T.P.; visualization, T.P.; supervision, M.L. and T.P.; project administration, M.L. and T.P.; funding acquisition, M.L. and T.P. All authors have read and agreed to the published version of the manuscript.

**Funding:** This research received no external funding.

**Institutional Review Board Statement:** Not applicable.

**Informed Consent Statement:** Not applicable.

**Data Availability Statement:** Data available on request from the authors.

**Conflicts of Interest:** The authors declare no conflict of interest.

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
