# Peer review of "Religious Self-Identification and Culture—About the Role of Religiosity in Cultural Participation"

_religions, doi:10.3390/rel12111028_

Round 1

Reviewer 1 Report

Page 2, Line 46: Difference between institutional religion and individual religion is not clear. Although the author mentioned the behavioral difference, the author needs to explain more about the difference.

P3, L 40: The definition of "cultural consumption" is not clear.  Can you take a example? For example, how Facebook treats religiosity?

P3. L 118: Although the author mentioned the difference between "religiosity" and "spirituality" later, I recommend the author to explain the difference here. 

P3. L135&136: Newman's definition that the author mentions is not collect. Although the author said, "Knowing (faith), Doing (religion) and Being (spiritual))," this logic is not always true. For instance, for Zen Buddhist and Yoga practitioner, "doing" is spiritual.

P4, L146: The researcher used the word, "inclusive." It is good. Then, I want to what is "exclusive." Can you explain about "exclusive"?

P4, L156&190: Religion might also be important for "sense of belonging to the community." Can you explain about the "sense of belonging" and "religious community"?

P5, L224: "Invisible religion" is not equal to the privatized religion. Some organized religion are also "invisible," such as online/net Churches.

P13, L452: Please explain more about the difference  between "religious person" and "spiritual person."

P13, L489: The definition of ""devoid of church" is not clear. Please explain more.

Author Response

Page 2, Line 46: Difference between institutional religion and individual religion is not clear. Although the author mentioned the behavioral difference, the author needs to explain more about the difference.

Thank you for your suggestion. We agree – the difference between institutional religion and individual religion was not clear in the text. We have made the necessary explanations.

P3, L 40: The definition of "cultural consumption" is not clear.  Can you take a example? For example, how Facebook treats religiosity? 

We do not fully understand the question. It seemed to us that we first of all had to define the concept we use - participation. The concept of consumption of culture was too closely related to the economy, which reduces culture primarily to a receptive attitude. Participation is something more – in the sense of being active, engaged, present and experiencing. But in the literature, however, these terms are sometimes understood as synonyms.

P3. L 118: Although the author mentioned the difference between "religiosity" and "spirituality" later, I recommend the author to explain the difference here. 

Thank you for this comment. We wondered how to change it, but after a few tries, we found that our corrections made changed the narrative and the logic of the argument. We decided, however, to leave this part of the text unchanged.

P3. L135&136: Newman's definition that the author mentions is not collect. Although the author said, "Knowing (faith), Doing (religion) and Being (spiritual))," this logic is not always true. For instance, for Zen Buddhist and Yoga practitioner, "doing" is spiritual. 

Thank you for this comment. It seems that this does not contradict the deification itself. It is true that Newman distinguished it in the analysis as two separate things (based on faith), but it is nowhere indicated in the text that it cannot interpenetrate. Being and doing can coexist in one activity also in other religions. But it doesn't have to. (One can do some practices even based on faith, but it doesn't have to be spiritual for this person.) Even Yoga can be practiced without spirituality. We have added an explanation in the text for the sake of clarity.

P4, L146: The researcher used the word, "inclusive." It is good. Then, I want to what is "exclusive." Can you explain about "exclusive"?

If we understand the reviewer's comment correctly, then "exclusive" would mean excluding from the analysis the aspects related to self-identification. What would be the consequences for the analysis? We wrote: “The inclusion of the aspect of spirituality and religiosity as self-identification is important because with the processes of secularization and desecuralization”. We think that the absence of these aspects in the analysis could mean a loss of sight of changes in contemporary religiosity. Which means staying at the level of the analysis of traditional, institutional religiosity, which - as we wrote in the paper - is looking for new forms to come into being. We have added a small addition.

P4, L156&190: Religion might also be important for "sense of belonging to the community." Can you explain about the "sense of belonging" and "religious community"? 

Belonging as identification with a group, a sense of belonging, describing yourself as "Catholic" is a good example here. Belonging to a community can also be very specific, related to direct social ties, network of contacts, etc. in a particulary religious community. We have added an explanation in the text for the sake of clarity.

P5, L224: "Invisible religion" is not equal to the privatized religion. Some organized religion are also "invisible," such as online/net Churches.

Of course, our mistake – thank you. Luckmann's understanding of "invisibility" is broader than we gave it in the paper - we have made a wrong simplification.We have added the necessary corrections.

P13, L452: Please explain more about the difference  between "religious person" and "spiritual person." 

We relied on self-identification of respondents here. A religious person, as a person who considers himself religious, identifies himself with a particular religion. A spiritual person as a person who considers himself a spiritual person and cares for his spiritual development. We have added an explanation in the text for the sake of clarity.

P13, L489: The definition of ""devoid of church" is not clear. Please explain more.

The term “devoid of church” was used (perhaps quite ineptly) as a metaphor (in meaning “non-church”). But the reviewer's remark is correct - this term may raise definitional doubts. We have abandoned its use in the text.

Reviewer 2 Report

The reviewed work is very good. It's been a long time since I read such thoroughly prepared, conducted and described research on a large research sample. I have no comments to the work, but I encourage the authors of the reviewed article to undertake a new task - related to a similar study, but during a COVID-19 pandemic. I wonder if people participating in religious life online also participated in online cultural events? Such a proposal for the future.

Author Response

Thank you very much for such a positive review. Thank you also for your suggestion for further research. The covid-19 pandemic definitely had an impact on cultural participation as well as on religious life, so it is a very interesting research topic.

This manuscript is a resubmission of an earlier submission. The following is a list of the peer review reports and author responses from that submission.

Round 1

Reviewer 1 Report

I find the paper on the role of religiosity in cultural participation very interesting. It is well designed and well presented.

In my opinion the following 3 issues have to be improved:

194-195 verbs in this sentence seems confusing or not gramatically correct

244 Whereas „family aspect” present in the hypotesis 2 is well desribed in section 2 and 3, the term „cultural heritage”, also included in the hipotesis H2, needs to be explained in the paper.  Why is it worth studying – generally speaking and in the case of Poland? I did not find any arguments supporting it.

Are there any new or inspiring issues which Authors would like to examine in future? In other words, Authors` opinion on future research on the topic of the role of religiosioty in cultural participation, put at the end of this research  paper, would add value to the paper and deepen the conclusions.

Reviewer 2 Report

Page 1, line 22: It was not clear the difference between religiosity, spirituality and culture.

Page 2, lines 52-57: It was not clear why and how Polish culture is unique. Are there any historical and/or empirical evidences? By the author's logic, every culture should be unique.

Page 2, lines 67-151: The author did not mentioned the religiosity and its dimensions of Polish culture.

Pages 3, line 153 - Page 5, line 248: The author did not mentioned the religiosity and Polish culture.

Page 5, line 250: Methods are okay, but there is no connection between the theories and methods.

Research questions and hypotheses are unclear because the theories are not based on the Polish "unique" culture.

Reviewer 3 Report

The article presented for review presents an interesting topic. However, the Author / Authors omitted important elements in their work that prevent the work from presenting a high scientific level. The article is improperly composed and thus gives the impression of being chaotic. I propose to use the traditional division of labor into: introduction, background, literature review, material and methods, results, discussion and conclusions.

  1. There is no description of the method used in the research. Line 63 is written, which is important, but it is not described how it will be done. I recommend extending the Materials and Methods point.
  2. Line 115 et seq. - treat as Background or Literature review.

Reviewer 4 Report

I propose to the Autor/Autores to expand the "Discussion" section to present the findings of other recent research on a similar topic. That last part of the article can be further improved.